# Implementing and Evaluating Community Health Worker-Led Cardiovascular Disease Risk Screening Intervention in Sub-Saharan Africa Communities: A Participatory Implementation Research Protocol

**DOI:** 10.3390/ijerph20010298

**Published:** 2022-12-24

**Authors:** Kufre Okop, Peter Delobelle, Estelle Victoria Lambert, Hailemichael Getachew, Rawleigh Howe, Kiya Kedir, Jean Berchmans Niyibizi, Charlotte Bavuma, Stephen Kasenda, Amelia C. Crampin, Abby C. King, Thandi Puoane, Naomi S. Levitt

**Affiliations:** 1Chronic Disease Initiative for Africa, Department of Medicine, University of Cape Town, Cape Town 7700, South Africa; 2Centre for Social Science Research, University of Cape Town, Cape Town 7700, South Africa; 3Department of Public Health, Vrije Universiteit Brussel, 1090 Brussel, Belgium; 4UCT Research Centre for Health through Physical Activity, Lifestyle and Sport, Division of Physiological Sciences, Department of Human Biology, Faculty of Health Sciences, University of Cape Town, Cape Town 7700, South Africa; 5Armauer Hansen Research Institute (AHRI), Addis Ababa P.O. Box 1005, Ethiopia; 6College of Medicine and Health Sciences, University of Rwanda, Kigali 4285, Rwanda; 7School of Medicine and Pharmacy, College of Medicine and Health Sciences, University of Rwanda, Kigali 4285, Rwanda; 8Malawi Epidemiology and Intervention Research Unit, Lilongwe P.O. Box 46, Malawi; 9Department of Epidemiology & Population Health, Stanford University School of Medicine, Stanford, CA 94305, USA; 10School of Public Health, University of the Western Cape, Cape Town 7535, South Africa

**Keywords:** cardiovascular diseases, citizen science, CVD risk, screening, referral, community health workers, COVID-19, implementation science

## Abstract

The increasing burden of non-communicable diseases (NCDs), particularly cardiovascular diseases (CVD) in low- and middle-income countries (LMICs) poses a considerable threat to public health. Community-driven CVD risk screening, referral and follow-up of those at high CVDs risk is essential to supporting early identification, treatment and secondary prevention of cardiovascular events such as stroke and myocardial infarction. This protocol describes a multi-country study that aims to implement and evaluate a community health worker (CHW)-led CVD risk screening programme to enhance referral linkages within the local primary care systems in sub-Saharan Africa (SSA), using a participatory implementation science approach. The study builds upon a prior community-driven multicentre study conducted by the Collaboration for Evidence-based Health Care and Public Health in Africa (CEBHA+). This is a participatory implementation research. The study will leverage on the CVD risk citizen science pilot studies conducted in the four selected CEBHA+ project countries (viz. Ethiopia, Rwanda, Malawi, and South Africa). Through planned engagements with communities and health system stakeholders, CHWs and lay health worker volunteers will be recruited and trained to screen and identify persons that are at high risk of CVD, provide referral services, and follow-up at designated community health clinics. In each country, we will use a multi-stage random sampling to select and then screen 1000 study participants aged 35–70 years from two communities (one rural and one urban). Screening will be done using a simple validated non-laboratory-based CVD risk assessment mobile application. The RE-AIM model will be used in evaluating the project implementation outcomes, including reach, fidelity, adoption and perceived effectiveness. Developing the capacities of CHWs and lay health worker volunteers in SSA to support population-based, non-invasive population-based CVD risk prevention has the potential to impact on early identification, treatment and secondary prevention of CVDs in often under-resourced communities. Using a participatory research approach to implementing mobile phone-based CHW-led CVD risk screening, referral and follow-up in SSA will provide the evidence needed to determine the effectiveness of CVD risk screening and the potential for scaling up in the wider region.

## 1. Introduction

Cardiovascular diseases (CVDs) are the leading cause of death and disability globally with over 17.9 million deaths in 2019, representing 32% of all deaths, and 38% of premature deaths from non-communicable diseases (NCDs) (1). Three quarters of these deaths occur in low- and middle-income countries (LMICs), where people often have limited access to primary care for early detection and treatment, resulting in delayed diagnoses and premature mortality [1,2,3]. The increasing burden of NCDs, particularly of CVDs in LMICs is a considerable threat to public health. The effects of CVDs on health systems have been exacerbated by the sustained COVID-19 pandemic [4], and there is a critical need for cost-effective and sustainable CVDs risk screening targeting the vulnerable population. This can be accomplished through task-shifting and or task-sharing, referral linkages, and mobile technology [5,6].

Community health workers (CHWs) have increasingly been engaged to address the human resource gaps as part of larger efforts to prevent, reduce and manage infectious diseases and NCDs [7,8]. However, CHWs often lack the training and tools to support early detection and management of those at high CVDs risk in resource-poor settings [9,10]. For instance, in Ethiopia, the national health extension programme (HEP) which is implemented at scale with a high coverage at community level, has led to the reduction in maternal and child health, communicable diseases (e.g., malaria case management), and immunization coverage, in turn helping to meet the Sustainable Development Goals [11]. However, there is no streamlined process yet to engage CHWs to implement CVD prevention strategies of the Ministry of Health (MOH). In Rwanda, CHWs played an important role in various health programmes such as maternal and child health, HIV, TB, malaria home-based care, vaccination campaigns and COVID-19 mitigation, as well as community-level education programmes [12,13]. Similarly, in Malawi, CHWs are involved in the delivery of integrated care and home-based care (including blood pressure monitoring, maternal and child health care, and vaccination campaigns) and not in CVD prevention [14]. In South Africa, the Department of Health has proposed the building of the capacity of CHWs in the management of chronic conditions through interprofessional collaboration and learning within a health care team that involves a professional nurse, a health promoter, and an environmental officer, who together form a primary health care outreach team [15]. Building the capacities of CHWs in SSA in population-based CVD prevention and referrals for care will expand the involvement and scope of practice of CHWs from primarily infectious disease management and MCH towards chronic care and prevention. This, in turn, is expected to increase the human resources pool needed for comprehensive and sustainable NCD prevention and care in the region.

While evidence-based data on the design and impacts of multi-setting participatory community-driven CVDs prevention project in SSA is limited, CHW-led CVDs risk screening and referral to care has been shown to be an important and effective strategy in helping tackle the burden of CVDs in LMICs [16,17]. In Nepal, a low-income country in South Asia, the effectiveness of using Community Health Volunteers in CVD risk screening have recently been reported [18].

This, according to the World Health Organisation (WHO), can help in improving the early containment of disease progression, and also address health workforce shortages and support universal health care delivery [7]. Effective screening, early identification and appropriate management of persons who are at high risk of CVDs can help create awareness for secondary prevention, supports early diagnoses, and facilitates reduction in CVDs in resource-poor settings [19]. Further, adopting a participatory implementation research approach will enhance planning and evaluation of the programme outcomes. We will use the RE-AIM framework to guide the study, as it appears the most consistent with our goal, that is, to reach the target populations, and assess the effectiveness of the CVD risk screening intervention uptake, and maintenance in care in the communities.

The study is part of the Collaboration for Evidence-based Health Care and Public Health in Africa (CEBHA+) project that aims at informing evidence-based policies on CVDs screening in Africa [20]. The CEBHA+ project teams identified four SSA countries, namely Ethiopia, Rwanda, Malawi, and South Africa to implement community-based CVDs risk prevention. In each of these four countries, citizen science pilot projects were conducted using trained citizen scientists to explore CVDs risk perceptions, and develop prevention and advocacy strategies to support CVDs risk communication and screening in the study communities [21,22]. The four selected project countries work in collaboration under CEBHA+ to support the implementation of community-based NCDs prevention to scale, focusing on CVD risk screening and referral to care. 

Studies have shown that CHWs in South Africa, Bangladesh and Mexico have accurately screened high-risk persons for CVD using a simple risk assessment tool, and conducted referrals and follow-ups at local clinics [9,23]. A recent global study showed the use of WHO non-laboratory based CVD risk score (versus laboratory-based scores) to predict CVD risk in 21 global regions [24]. Our study intends to build on this, adding the use of mobile-phone technology, and participatory implementation research to support CVD risk screening, referral, and follow-up in resource-limited settings so as to enhance the effective uptake of screening and referral services [25]. An implementation research approach is employed to facilitate the planning and evaluation of the implementation and outcomes in terms of reach, implementation fidelity and cost, adoption, and maintenance [26,27] within the local PHC systems in SSA communities. The main objective of this study is to implement and evaluate CHW-led population-based CVD risk screening using an implementation research approach to assess CVD risk screening and referral outcomes in the designated countries.

## 2. Materials and Methods

***Design:*** This study will utilize a participatory implementation science approach. The schema for study design, including the data collection and outcomes evaluation plan is presented in Figure 1. 

***Setting and study population:*** The study will be conducted in the rural and urban communities of four African countries in the CEBHA+ network, namely Ethiopia, Rwanda, Malawi and South Africa. In each study community, two designated local community health centres (CHCs) will be selected by the community through community engagement processes during the citizen science pilots. Study participants will be recruited from the catchment areas served by selected CHCs in each of the participating sites.

***Sampling and Sample Size:*** We will use a multi-stage sampling technique to select participants. The study community (rural or urban) will be chosen among the communities within the selected catchment area in each country. From a designated central point in the chosen study community, the second compound (or household) will be selected and one or two participants selected for screening until the sample size is reached. A sample size of 1000 (or minimum 900) study participants per country was calculated based on the following assumptions: we assumed that 3% or 4% of the study populations would have a CVD risk score > 20% based on a previous CVD risk screening study that reported a proportion of 5–6% in four LMICs (Bangladesh, Mexico, Guatemala and South Africa) [28]. We also assumed that to be able to assess referral outcomes, we need 30 to 40 persons per country who are at high CVD risk (i.e., risk score ≥ 20%) to be referred for care and followed up. To realize 40 eligible persons for referral and follow-up, we need a sample size of 1000 (i.e., 4 in every 100 persons surveyed) per site. To release at least 30 at risk persons, a minimum sample size of 900 will be required. This sample will allow comparisons among the different study sites participating in the study, as well as a multivariable analysis adjusted for potential confounding factors. 

***Eligibility Criteria:*** Men and women aged 35–70 years who live in the selected communities and are available for screening are eligible for the study. Those with a known history of CVD (e.g., stroke, myocardial infarction, heart attack, heart failure, angina, etc.) will be excluded, but provided with counselling and referral services. During recruitment, we will ascertain if the potential study participants had been diagnosed of any known common CVDs, by probing and seeking confirmation from close relatives.

***Study Scope:*** Trained CHWs will use a simple non-laboratory-based Framingham CVD risk assessment algorithm programmed in smart mobile phones/tablets to collect data, screen and identify persons who are at high risk of CVD. The CHWs will then refer the at-risk persons to community health clinics, and follow them up to ascertain the care they received. Findings of the CVD risk citizen science qualitative pilots conducted recently in the four selected CEBHA+ project countries (viz. Ethiopia, Rwanda, Malawi and South Africa) that provided information on CVD risk perception, interpretation and communication of health related risk will be utilised to inform an effective implementation of CVD risk screening in the study communities [22,29,30]. The expected duration for the study is 12–18 months.

***Intervention Delivery****:* The intervention will be conducted in four successive phases viz. (i) Citizen science pilot; (ii) Recruitment and training of CHWs and CHVs; (iii) CVD risk screening and referral; (iv) Follow-up and evaluation.

(i)
*Citizen science pilot:*


Citizen science pilot study has already been conducted in three countries and is ongoing in the fourth country (as at September 2022) to explore community-level CVD risk perceptions. This step is needed to inform the development of a training curriculum in each project country and settings.

(ii)
*Recruitment and training of CHWs:*


Four key criteria will be used for recruiting the CHWs. These include (i) currently serving as a CHW or CHEW or CHV in designated project communities; (ii) having completed CVD risk training and scored at least 75% on evaluation test; (iii) have confirmed availability (for at least two consecutive months) during the project implementation; and (iv) are committed and willing to participate in this study.

A total of 24 CHWs/CHVs will be recruited in each country; 12 CHWs each from a rural and an urban community. Two CHW coordinators (one each from a rural and urban site) will be trained in each country to support the supervision of the CHWs during the study. CHWs/CHVs will be trained to obtain anthropometric and risk factor data, including medical history, and to enter these data into the CVD risk mobile app to generate an absolute CVD risk score. A 5–6-day training will be conducted, including both didactic and practical components for undertaking CVD risk assessment. Didactic training will cover definition and good characteristics of research, research ethics, CVD risk definitions, symptoms and risk factor history assessment, and use of the mobile data collection and CVD risk screening app. The training evaluation will consist of a pre- and post-training knowledge test, standardized anthropomorphic measurement skills and blood pressure measurement evaluation, and a risk score reading on the mobile app. Each CHW is expected to score up to 75% of the average scores to be qualified to undertake community screening.

Training curriculum: The training curriculum for CVD risk screening and referral used in our previous study in South Africa will be adapted for use in the countries [31]. The curriculum will incorporate lessons learnt in each country from the citizen science project and the qualitative research on CVD risk perception and communication as part of the CEBHA+ project conducted between 2019 and 2022 [22,30]. The curriculum comprises three modules: *Module 1* (Basics of CVD and community perceptions) aims to explore what CHWs/CHVs know and think about CVD and its associated risk factors and will inform trainees about how members of their community perceive CVD risk, its causes and the importance of screening. *Modules 2* and *3* cover practical skills required for conducting the CVD risk assessment and the evaluation process of the screening. 

(iii)
*Population-based CVD risk screening and referral linkages:*


In each project community, the two CHCs will serve as a hub for this project. CHWs will conduct screening, referral and follow-up with the assistance of a mobile app, and will be based on the risk screening and referral protocol. A minimum of 10 trained CHWs/CHVs per study site will conduct a risk assessment survey to screen participants in their homes over the course of 6–8 weeks. Each CHW/CHV will be expected to screen not more than five persons per day, until he/she screens a total of 100 persons. Informed consent for participation will be obtained prior to CVD risk screening. Screened participants will be offered counselling, and those at high risk will be provided referral services, and followed-up thereafter by the CHWs. Prior to commencement of the fieldwork, the research team in each country will liaise with their respective community health centre managers to discuss steps to support referral and linkage to care for the at-risk persons referred by the CHWs. Referral linkages and support will be developed through consultation with relevant stakeholders in primary health care (PHC) including the Department of Health (DoH), Ministry of Health (MoH) personnel. Not less than 10 PHC staff (managers, nurses, social workers) and representatives from the MOH, DoH in the respective countries will be involved in project planning, training and implementation. This will facilitate acceptance at the clinic, and follow-up of patients referred to the health care system. Referral forms and registers developed for the previous CHW-led screening pilot in South Africa will be adapted for this study [23]. 

(iv)
*CVD risk screening and referral procedures:*


The screening, referral and follow-up will be done using a CVD risk assessment tool (on a mobile phone). This tool was tested and validated with a good performance in several South African cohorts, and also used to screen persons in Bangladesh, Guatemala and Mexico [23,31,32]. The CVD risk assessment tool uses the variables of age, gender, BMI (instead of lipids), systolic blood pressure, smoking status, and diagnosed diabetes and hypertension to calculate an absolute risk score for developing CVD [33]. Our most recent country-level analysis of national NCD population-level data (based on WHO STEPS surveys) indicated a high level of comparability between non-laboratory-based and laboratory-based CVD risk scores in the reference populations for Rwanda [34] and other countries in the SSA region [24]. The procedure for screening that will be used (as a standard operating procedure) by the CHWs/CHVs is given in Section A.1. To explore more about CVD risk a set of short questions on personal information, risk perception and health-seeking behaviour will be asked using the mobile-app questionnaire (See Section A.2). The referral commences once a client is screened and is shown to be or high risk (i.e., risk score ≥ 20). CHWs initiate the referral using her mobile phone app, and laisses with the nurses in the CHCs for booking of clients for the risk assessment in the clinic (see Figure 1)

(v)
*Follow-up and evaluation of CVD risk screening and referral outcomes:*


Persons identified and referred to the clinic will be followed up by CHWs/CHVs within eight weeks (2 months) post screening. The compilation of referral and follow-up outcomes will be undertaken by CHWs/CHVs, and data entered in the mobile phone app and synchronized. Qualitative interviews will also be conducted with the beneficiaries and the implementers to provide information on reach, acceptability, effectiveness and cost. There will be no monetary compensation for participating in this study, as screening, counselling, referral and follow-up services will be provided.

We will use the RE-AIM framework to facilitate our assessment of the *reach* (proportion of at-risk persons who attended clinic and received/did not receive care), *effectiveness* (numbers screened, referred and followed-up, and perceived satisfaction), *adoption* (use of screening and referral processes), *implementation* fidelity, and *maintenance* in care (sustainability of programme—screening and referral) [35]. The RE-AIM framework is suitable, as it would facilitate the planning, and the evaluation of the implementation outputs and outcomes that will enable us to know how best to implement a cost-effective and sustainable CVD risk screening intervention at population-level in SSA.

***Data Collection and management:*** Participants’ contact information (including phone numbers, address, study ID numbers, geo-code (location), age, sex, history of smoking, history of stroke, heart attack, diabetes, hypertension, and treatment) will be collected with the mobile app. Anthropometric measures (weight, height, waist measurement), blood pressure (systolic, and diastolic—SBP and DBP) will be used to calculate an individual risk score. Persons screened and identified as ‘high-risk’ (i.e., 10-year risk score ≥ 20) will be referred to the designated CHC. Data will be aggregated and archived in a secure server in each country in designated research project offices. All data will be held in encrypted repositories and will be curated to enable an effective analysis and comparison across countries. Data sharing agreements and country-level procedures for safeguarding data will be adhered to. Individual sites will complete the necessary procedures to ensure compliance with their respective institutional review boards for guaranteeing the protection of confidentiality of study participants during the study, including recruitment and follow-up. The aggregated data will be shared for study-wide analyses and dissemination.

*Data Analysis and evaluation**:*** Data from all participating countries will be harnessed for analysis. Results will be stratified by individual sites to identify trends or between-site differences. Country-level analysis and results dissemination will also be undertaken. Key expected outcomes include CHW/CHV knowledge retention, number of persons screened and provided counselling, number identified as high-risk and patients referred for care, number of patients attended to at CHCs, and follow-up and care outcomes. ANOVA tests will be used to assess if there is a significant difference in the number of patients referred and attended clinics, by gender and location. Furthermore, we will assess the risk score, risk perception and health-seeking behaviour between country sites. 

Using the RE-AIM model, we will ensure that reach (proportion of at-risk persons referred, attended a clinic and received/not received), adoption (use of screening and referral processes), perceived effectiveness (beneficiaries’ satisfaction, those maintained in care (within first 2–3 months)) and implementation fidelity (adherence to steps including counselling and referrals) will be assessed.

The evaluation of the screening and referral outcomes will be done by comparing set indicators described in the methods section at baseline and follow-up. The indicators for evaluation are outlined in Table 1. Key amongst these include: (i) proportion of persons screened that are at high-risk for CVDs; (ii) number referred for care; (iii) proportion of high-risk persons obtaining further assessment, diagnosis and treatment; (iv) proportion of at-risk persons receiving a follow-up visit by CHWs/CHVs after diagnosis or treatment; and (v) participant’s self-evaluation of satisfaction of care at the clinic from referral to care utilization. The mean costs for training and implementing the screening programme overall will also be evaluated and compared across study project countries.

## 3. Discussion

This protocol paper describes a multi-country population-based CVD risk screening and referral study to identify individuals at high risk of CVD for referral into care in poor SSA rural and urban communities. This initiative will not only enhance CHW-led non-invasive and simple CVD risk screening but will support and strengthen referral linkages to care for persons with NCDs (especially CVDs) in poor communities. These efforts will provide access to screening and care for persons living in under-served and economically disadvantaged communities who often do not have access to NCD care. Implementing and evaluating a sustainable community-driven CVDs risk screening and referral to primary health care facilities in SSA communities will support evidence-based strategies for implementing cost-efficient secondary prevention programmes [28] to mitigate the rising burden of NCD in such settings. 

This study will also facilitate community participation and capacity strengthening in research planning, implementation and evaluation in four SSA countries using implementation research. We envisaged that in this project, approximately 9500 persons (including community members, stakeholders, and researchers) will be impacted through training, advocacy workshops and research engagements in the eight selected communities in the four project countries. This human capacity development, and community-driven learning and advocacy efforts towards CVD prevention is of critical public health importance, especially as it is essential to address the rising CVD burden in the vulnerable populations in SSA. Implementation research has clear relevance to addressing the rising burden of NCDs in SSA, especially CVDs, as it facilitates multi-level participation and increases the potential for scalability of the intervention for wider implementation in the SSA region.

Recent systematic reviews have shown the value of implementation research to guide the implementation of NCDs prevention and control in LMICs [36,37]. It is also evident that utilising integrated knowledge translation (IKT) strategies within implementation research can support effective engagement in project implementation, evaluation and dissemination [22,38]. Therefore, we envisaged that engaging with communities, stakeholders and advocates using a participatory approach such as citizen science and IKT strategies will help support community-driven multi-country population-based CVD risk screening and referral towards achieving CVD prevention and management in Africa.

## 4. Conclusions

Implementing a population-based CVDs risk screening, referral and follow-up led by CHWs and CHVs will expand the involvement of CHWs in NCDs and chronic care prevention. Overall, the intervention, if successful, has the potential to help mitigate the burden of CVDs in the underserved and vulnerable populations in the project countries. Using implementation research will help to assess intervention reach, feasibility, effectiveness, adoption, implementation fidelity, cost and sustainability of the intervention, as well as assess its potential for scaling up in the wider region.

## Figures and Tables

**Figure 1 ijerph-20-00298-f001:**
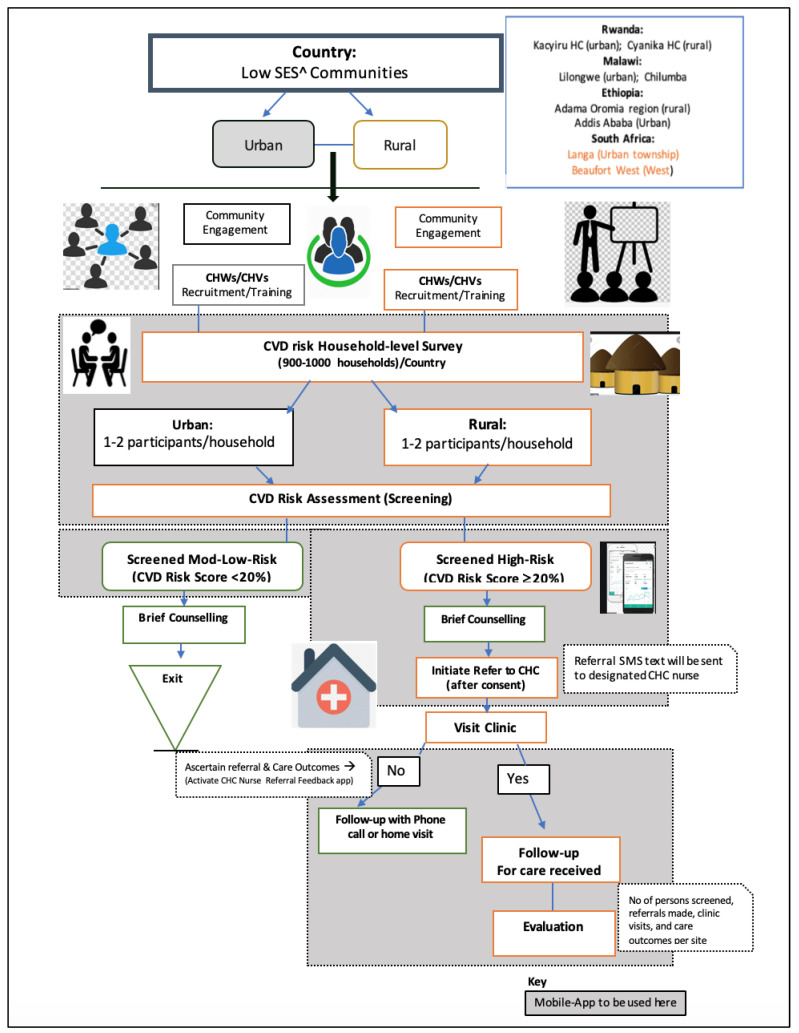
Schema for the study design.

**Table 1 ijerph-20-00298-t001:** Indicators to be measured (by RE-AIM dimensions).

RE-AIM Dimensions	Indicators	Data Collection (Means of Verification)
Reach	− Proportion and representativeness of at high-risk persons and referred for care; proportion that were attended to at clinic	Quantitative Survey
Effectiveness *	− Proportion of high-risk persons that attended a clinic and received/not received further assessment − Perceived satisfaction (self-evaluation of satisfaction from screening, referral to care utilization)	Referral and follow-up key informant interview Facility level data Qualitative interviews
Adoption	− Number proportion of communities/settings willing to initiate screening programme − Reasons for adoption or non-adoption	Key informant interviews/FGDs
Implementation	− Implementation fidelity (adherence to steps including screening, counselling and referral protocol) delivery time required − Adaptations made (to processes/tools)	Programme report
Maintenance	− Proportion of communities/settings continuing to screen for CVD risk prevention using CHWs after a set interval or after the study has ended	Key informant interviews/ observation checklists

Cost	− Costs of implementation (mean cost of training, and implementing screening programme in each project site/country)	Budget/Finance and reports systems
Acceptability/feasibility	− Acceptability in terms of satisfaction of service use by beneficiaries.	Follow-up observation checklists and qualitative interviews

* Individual level outcome or quality of life will not be measured—as these are outside the scope of this study.

## Data Availability

This is not applicable.

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
