# Peer review of "Implementing and Evaluating Community Health Worker-Led Cardiovascular Disease Risk Screening Intervention in Sub-Saharan Africa Communities: A Participatory Implementation Research Protocol"

_ijerph, 2022, doi:10.3390/ijerph20010298_

Round 1

Reviewer 1 Report

I would like to congratulate the authors on this important aspect of preventive cardiology and identifying high-risk patients to provide interventions. Also, it is impressive the methodologies that have been used. 

Author Response

We the authors thank you for this important positive comment.

Reviewer 2 Report

1.       The link for Ref3 to be corrected https://ncdalliance.org/resources/invest-to-protect-ncd-financing-as-the-foundation-for-healthy-societies-and-economies

2.       Line 77-80. I suggest authors adding a reference which showed effectiveness of using Community Health Volunteers in a Low-Income country in South Asia: Rawal, Lal B., et al. "Engaging Female Community Health Volunteers (FCHVs) for cardiovascular diseases risk screening in Nepal." Plos one 17.1 (2022): e0261518.

3.       Lines 128-130, Lines 158-159, Lines 238-241. I suggest adding a reference which has specified cardiovascular disease risk laboratory-based charts for 21-subregions (please see appendix 2). I think considering these charts would add value where interpreting results. Kaptoge, Stephen, et al. World Health Organization cardiovascular disease risk charts: revised models to estimate risk in 21 global regions." The Lancet Global Health 7.10 (2019): e1332-e1345.

4.       Lines 192-193. CHW Recruitment criteria, I suggest authors make the criteria more inclusive and simpler i.e. Rawal et al (2022) did: Four key criteria were used for recruiting FCHVs, including (i) working as a FCHV in the study sites; (ii) having completed at least 8th grade of education or above; (iii) older than 18 years old; and (iv) voluntary willingness to participate in this study.

5.       Line 247, the reference should be 29 not (in press)

6.       Figure 1. Sample size 900-1000 households/country, whereas 1000 is mentioned in the text throughout the manuscript. I would suggest consistency.

7.       Lines 437-439. Appendix 2. Q9. The options for willingness for heart check up ranging from 0 to 5 seems unreasonable: it should be either “Yes” or “No”. If the authors want to add additional category of “Undecided”

8.       Additional Comment: Since this is a multi-country/multi-centre study the study must be applying a range of filed activity monitoring and supervision. Therefore, I would suggest the authors to add a section on Study Governance.

Author Response

Comment #1: The link for Ref3 to be corrected https://ncdalliance.org/resources/invest-to-protect-ncd-financing-as-the-foundation-for-healthy-societies-and-economies

Response: Thank you for providing the link. We have added it to the reference.

Comment #2: Line 77-80. I suggest authors adding a reference which showed effectiveness of using Community Health Volunteers in a Low-Income country in South Asia: Rawal, Lal B., et al. "Engaging Female Community Health Volunteers (FCHVs) for cardiovascular diseases risk screening in Nepal." Plos one 17.1 (2022): e0261518.

Response: Thank you for this excellent and most recent reference. We have made a clear statement about the example of engaging CHVs to support CVD risk screening in Nepal, a Low income country in Asia. We have also added this to the reference (see reference #18).

      Comment #3: Lines 128-130, Lines 158-159, Lines 238-241. I suggest adding a reference which has specified cardiovascular disease risk laboratory-based charts for 21-subregions (please see appendix 2). I think considering these charts would add value where interpreting results. Kaptoge, Stephen, et al. World Health Organization cardiovascular disease risk charts: revised models to estimate risk in 21 global regions." The Lancet Global Health 7.10 (2019): e1332-e1345.

Response: Thank you for this excellent and most recent reference. We have made a clear statement about the WHO global study on CVD risk prediction charts – see lines 129-133 in the Introduction Section. We have also added this reference to the manuscript reference (see reference #24).

 Comment #4: Lines 192-193. CHW Recruitment criteria, I suggest authors make the criteria more inclusive and simpler i.e. Rawal et al (2022) did: Four key criteria were used for recruiting FCHVs, including (i) working as a FCHV in the study sites; (ii) having completed at least 8th grade of education or above; (iii) older than 18 years old; and (iv) voluntary willingness to participate in this study.

Response: Thank you for this critically important comments and excellent inputs. We have now clearly stated the key inclusion criteria for the CHWs on page 4, lines 295-299. This statement is added under the Recruitment and training of CHWs sub-section (page 4) as follows:

:

Four key criteria will be used for recruiting the CHWs. This include (i) currently serving as a CHW or CHEW or CHV in designated project communities; (ii) having completed CVD risk training and scored at least 75% on evaluation test; (iii) have confirmed availability (for at least two consecutive months) during the project implementation; iv) committed and willing to participate in this study.

Comment #5: Line 247, the reference should be 29 not (in press)

Response: Thank you for this important comment and inputs. We have added the reference of the WHO global study on CVD risk prediction charts, which is much appropriate for the points were were making on comparability of CVD risk scores in the LMICs. The reference that fit is the current reference #2, that is, Kaptoge, Stephen, et al. World Health Organization cardiovascular disease risk charts: revised models to estimate risk in 21 global regions." The Lancet Global Health 7.10 (2019): e1332-e1345.

Comment #6: Figure 1. Sample size 900-1000 households/country, whereas 1000 is mentioned in the text throughout the manuscript. I would suggest consistency.

Response: Thank you for this important input. We have made a clarification on this. A maximum sample of 1000 or minimum sample of 900 were derived for our study. The clarifications are given under Sampling and Sample Size sub-section – on lines 154-266.

Comment #7: Lines 437-439. Appendix 2. Q9. The options for willingness for heart check up ranging from 0 to 5 seems unreasonable: it should be either “Yes” or “No”. If the authors want to add additional category of “Undecided”

Response: Thank you for this important comment and inputs. I have corrected the question #9 on the appendix 2 as advised.

Comment #8: Additional Comment: Since this is a multi-country/multi-centre study the study must be applying a range of filed activity monitoring and supervision. Therefore, I would suggest the authors to add a section on Study Governance.

Response: Thank you for this important comment. The Study Governance for this study is embodied in the global CEBHA+ ethics, and engagement guidelines/policies that were developed to guide our the project coordination throughout the period of the project in the countries at all levels. We have mentioned about the CEBHA projects goals and scope in the introduction and the methods sections. However, we do not think we need to give a separate section to describe Study Governance for our research network

Reviewer 3 Report

The objective of this protocol is to implement a community health worker-led cardiovascular disease (CVD) risk screening programme to 1. train health workers to identify and screen high CVD risk individuals using a non-laboratory based Framingham algorithm programmed in a smart phone. 2.Individuals at high risk of CVD will be referred to clinics and followed-up within 2 months post screening. Implementation research on CVD program delivery can illuminate what does and does not work in achieving CVD control.

Some comments regarding the structure of this protocol are as follows:

The introduction part described the increasing burden of cardiovascular disease and it's critical for the community health workers get more training and tools to support early detection of CVD. Paragraph 2 and paragraph 5 might be condensed together to describe the important role of community health workers in Rwanda, Malawi, Ethiopia and South Africa. The introduction could be much concise.

Access to care: It’s important to note that healthcare systems in sub-Saharan Africa communities are designed to take care of acute communicable diseases rather than the preventable NCDs. The referral procedures need more details to guarantee the rural patient attendance such as how many clinics are available in this community and how far is it, does the clinic have established protocol for CVD intervention etc. these aspects need to be considered.

Eligibility Criteria needs to be more specified: Those with a known history of CVD will be excluded. Is this CVD term including all the disease listed by world Health Organization?

The discussion section mentioned using integrated knowledge translation (IKT) strategies, IKT defined as an ongoing relationship between researchers and decision-makers (clinicians, managers, policymakers, etc.) for the purpose of engaging in a mutually beneficial research project or program of research to support decision-making. Explain this term may help readers better understand this paragraph.

Reginal collaboration, the culture of sub-Saharan African populations is encouraged to be discussed and how this protocol compared with previous working model?

Author Response

Comment #1: The objective of this protocol is to implement a community health worker-led cardiovascular disease (CVD) risk screening programme to 1. train health workers to identify and screen high CVD risk individuals using a non-laboratory based Framingham algorithm programmed in a smart phone. 2.Individuals at high risk of CVD will be referred to clinics and followed-up within 2 months post screening. Implementation research on CVD program delivery can illuminate what does and does not work in achieving CVD control.

Response: Thank you for this important positive comment.

Some comments regarding the structure of this protocol are as follows:

Comment #2: The introduction part described the increasing burden of cardiovascular disease and it's critical for the community health workers get more training and tools to support early detection of CVD. Paragraph 2 and paragraph 5 might be condensed together to describe the important role of community health workers in Rwanda, Malawi, Ethiopia and South Africa. The introduction could be much concise.

Response: Thank you for your helpful comments and suggestions. The authors have combined the paragraphs 2 and 5, condensed them, to describe the important role of CHWs in Rwanda, Malawi, Ethiopia and South Africa, and why CHWs-led CVD risk is needed. See the new paragraphs 2 – lines 67-90.

Comment #3: Access to care: It’s important to note that healthcare systems in sub-Saharan Africa communities are designed to take care of acute communicable diseases rather than the preventable NCDs. The referral procedures need more details to guarantee the rural patient attendance such as how many clinics are available in this community and how far is it, does the clinic have established protocol for CVD intervention etc. these aspects need to be considered.

Response: Thank you for this important comment and suggestion. We have addressed these important point in the Intervention Delivery sub-section (lines 282-359) and also in the detailed Protocol for recruitment, screening, and referral (Appendix 1). Specifically, the selection of community health clinics (CHCs), number per study sites (i.e. 2 per site), the recruitment and screening procedures, and referral linkages are also outlined in the following aspect of the methods sections ii) Recruitment and training of CHWs; iii) Population-based CVD risk screening and referral linkages, and iv) Follow-up and evaluation of CVD risk screening and referral outcomes – lines 282-359.

Comment #4: Eligibility Criteria needs to be more specified: Those with a known history of CVD will be excluded. Is this CVD term including all the disease listed by world Health Organization?

Response: Thank you for this excellent comment. The authors have further elaborate on the eligibility criteria (line 257-261). This is stated as follows.

Eligibility Criteria: Men and women aged 35-70 years who live in the selected communities and are available for screening are eligible for the study. Those with a known history of CVD (e.g. stroke, myocardial infarction, heart attack, heart failure or angina, etc) will be excluded, but provided with counselling and referral services. During recruitment, we will ascertain if the potential study participants had been diagnosed of any known common CVDs, by probing, and seeking confirmation from close relative.

Comment #5: The discussion section mentioned using integrated knowledge translation (IKT) strategies, IKT defined as an ongoing relationship between researchers and decision-makers (clinicians, managers, policymakers, etc.) for the purpose of engaging in a mutually beneficial research project or program of research to support decision-making. Explain this term may help readers better understand this paragraph.

Response: We appreciate this important comment, and your suggestion. The authors have revised the paragraph on important of using integrated knowledge translation (IKT) strategies to support community-driven intervention to prevent CVD. Kindly see this revision on lines 515-521, in the Discussion section of the paper.

Comment #6: Regional collaboration, the culture of sub-Saharan African populations is encouraged to be discussed and how this protocol compared with previous working model?

Response: We thank you for these very comments on regional collaboration, the culture of African populations, and how our protocol is set in light of other models of prevention (if I might I can understand this clearly).

Based on this comments, we have modified some of the statements in our introduction. As stated in the introduction, this study is part to a larger study facilitated under the Collaboration for Evidence-based Health Care and Public Health in Africa (CEBHA+) research network.

We have articulated the collaborative work between the project countries under CEBHA+, which is aimed at developing and implementing NCDs/CVDs prevention and care interventions in sub-Saharan Africa. Kindly see lines 117-126 under the introduction for these details.
